# Optimization of Key Technologies for Induction of Embryogenic Callus and Maturation of Somatic Embryos in Korean Pine (*Pinus koraiensis*)

Fang Gao [1,2], Yujie Shi [1], Ruirui Wang [1], Iraida Nikolaevna Tretyakova [3], Alexander Mikhaylovich Nosov [4,5], Hailong Shen [1] and Ling Yang [1,*]

1    State Key Laboratory of Tree Genetics and Breeding, School of Forestry, Northeast Forestry University, Harbin 150040, China; fanggao@nefu.edu.cn (F.G.); nefu_wrr@163.com (R.W.); shenhl-cf@nefu.edu.cn (H.S.)
2    School of Mine Safety, North China Institute of Science Technology, Langfang 065200, China
3    Laboratory of Forest Genetics and Breeding, Institution of the Russian Academy of Sciences V.N. Sukachev Institute of Forest Siberian Branch of RAS, Krasnoyarsk 660036, Russia
4    Department of Cell Biology, Institute of Plant Physiology K.A. Timiryazev, Russian Academy of Sciences, Moscow 127276, Russia
5    Department of Plant Physiology, Biological Faculty, Lomonosov Moscow State University, Moscow 119991, Russia
*    Correspondence: yangl-cf@nefu.edu.cn

**Abstract:** Somatic embryogenesis (SE), which leads to the formation of embryonic callus (EC) tissue, is the most promising method for large-scale production and selective breeding of woody plants. However, in many species, SE suffers from low induction and proliferation rates, hindering the production of improved plant materials. We investigated the effects of the explant sterilization method, 4 °C cryopreservation, basal medium, ethylene removal, liquid medium supplementation, and a combination of PGRs on embryogenic callus (EC) induction of Korean pine, using immature embryos of Korean pine as explants. The effects of sucrose and maltose on EC proliferation and maturation were investigated. The differences in the maturation ability of EC somatic embryos before and after cryopreservation were evaluated using the induced embryonic cell lines. The results showed that zygotic embryos (ZEs) performed better than megagametophytes (MGs) as explants. The induction rate of EC was significantly increased after 28 days of cryopreservation at 4 °C. The induction rate of EC in the #5 family increased from 10.00% to 62.8%. The EC induction rate of the five families cultured with the DCR basal medium was higher than that with the mLV basal medium. Among them, the induction rate of the #5 family cultured with the mLV basal medium was 23.3%, while that with the DCR basal medium was 60.9%, an increase of 2.6 times. There was no significant difference in the maturation ability of EC somatic embryos before and after cryopreservation. In conclusion, this study provides a method to improve the EC induction rate and maturation ability of Korean pine.

**Keywords:** Korean pine; embryogenic callus; maturation; explant selection; cryopreservation; carbon source





## 1. Introduction

At present, Korean pine (*Pinus koraiensis* Sieb. et Zucc; Pinaceae) is vastly distributed in the eastern mountains of northeast China (Changbai Mountains and Lesser Khingan Mountains), the far east of Russia, and the Korean Peninsula. There is also sporadic distribution in the alpine mountains in Honshu, Japan. Korean pine has a strong cold tolerance and long lifespan, and its age can reach more than 500 years [1]. The wood of Korean pine is excellent, and has been used as high-quality materials for buildings, bridges, furniture, flooring, etc. Korean pine is also an important greening tree species for landscaping and urban green spaces. Therefore, Korean pine has important economic

and ecological values. At present, there are many reports on the general bioecological characteristics, growth and development, breeding, seedling afforestation, and community dynamics and structure of Korean pine [2–4]. However, there are few reports on the tissue culturing of Korean pine. Our previous studies showed that the immature zygotic embryo (ZE) of Korean pine as explants could be induced to embryogenic calluses (ECs), but the induction rate of EC was significantly affected by genotype, and the EC gradually lost its embryogenic potential over time, which seriously affected the formation and yield of somatic embryos [5–8].

Explant sterilization is an essential part of somatic embryogenesis (SE). Alcohol and sodium hypochlorite are commonly used for sterilization in conifer SE [9]. Reeves et al. [10] have shown that sterilization of explants could significantly affect the induction rate of EC, and that sterilization of intact immature cones would be sufficient to prevent contamination of the culture after inoculation, and could also increase the induction rate of EC. However, the sterilization methods for different tree species, different degrees of maturity, and different explants are different, which need to be determined through experiments. The explant is another key to the success of EC induction [11], which depends on the genotype and the type of explant. The effect of the embryo's developmental status (maturity) on EC formation and SE is quite different [12,13]. The EC induction with mature embryos is more difficult than with immature embryos [14]. With the increase of the mother tree's age and the change in physiological state, the induction rate of EC will decrease significantly [15,16]. Different genotypes have significant effects on EC induction rate [17,18], increasing the difficulty of large-scale reproduction. ZEs attached to the kernel were reported to be the best explant for EC induction in most studies [19,20]. The reason is that ZEs will absorb more growth regulators and nutrients from the embryo cavity [10,21], which will contribute to the initiation of embryonic cells. Reeves et al. [10] showed that the induction rate of EC with ZEs as explants was significantly higher than that with megagametophytes (MGs) as explants. The induction rate of ECs using MGs as explants was very low, and the induction rate of ECs using ZEs as explants increased from 16% to 55%. The difference in EC induction between ZEs and microsporangia of Korean pine as explants is not clear, and further studies are needed to determine the optimal explants. Sugar is not only a kind of osmoregulatory substance, but also a carbon and energy source for the medium. In *Abies fabri* SE, the endogenous carbohydrate pattern is stable regardless of culture conditions, indicating that the carbohydrate status is an important feature of the normal development of *Abies fabri* SE [22]. Plant growth regulators (PGRs) are considered to be the most important factors in the process of EC induction, proliferation, and differentiation [23–25], and play an important role in regulating the whole SE process [23]. 2,4-dichlorophenoxyacetic acid (2,4-D) and α-naphthalene acetic acid (NAA) are the most commonly used auxins. 6-benzylaminopurine (6-BA) and kinetin (KT) are the most commonly used cytokinins. However, different tree species have different responses to PGRs. Therefore, the appropriate concentrations and specific combinations of PRGs need to be screened. In addition, the embryonic maintenance and somatic embryo yield of EC is an important evaluation index in the large-scale breeding of conifer SE. Somatic embryos and regenerated plants can be obtained from EC of Korean pine after cryopreservation and resuscitation [7]. However, the difference in maturation ability before and after resuscitation of different genotypes and materials has not been systematically evaluated.

This study was divided into three main research areas, with an overall objective to overcome the main limitations in commercializing Korean pine SE technology. Specifically, the first goal was to improve EC initiation efficiency. The second goal was to screen for the conditions suitable for proliferation by modifying the proliferation medium and culture density in culture. The last goal was to evaluate effects of cryopreservation on somatic embryo yield of different genotypes of embryonic cells. We expect that this study will help the forest sector overcome the barriers and develop feasible and productive protocols for mass vegetative propagation of Korean pine.

## 2. Materials and Methods

### 2.1. Plant Material

The collection of plant materials in this study conforms to international standards. The collection of all materials was authorized by the cooperative institution of Lushuihe Seed Orchard, Jilin Province, China.

On 1 July 2018, the cones of 7 full sibling families (Numbered #1 (female parent * male parent: 176 * 174), #2 (female parent * male parent: 166 * 158), #3 (female parent * male parent: 176 * 175), #4 (female parent * male parent: 174 * 174), #5 (female parent * male parent: 175 * 161), #6 (female parent * male parent: 166 * 161), and#7 (female parent * male parent: 174 * 158) of Korean pine were obtained from Lushuihe Seed Orchard, Jilin Province, China.

On 1 July 2019, open-pollinated plus mother tree cones of 3 Korean pine families (numbered #8, #9, and #10) were obtained from the Lushuihe Seed Orchard, Jilin Province, China.

On 13 July 2020, open-pollinated cones of 3 families (numbered #11, #12, and #13) of Korean pine were obtained from the Lushuihe Seed Orchard, Jilin Province, China.

### 2.2. EC Induction of Korean Pine

#### 2.2.1. Sterilization Method of Cones

The MG of the #8, #9 and #10 families were used as explants. Sterilization method: The Korean pine cones were washed with detergent for about 30 min, and then with running water for 8 h. Method 1: The cones were sterilized in 75% alcohol for 50 min on a super-clean bench. The seeds were collected 1 h after the cones were sterilized. The seeds were soaked in 75% alcohol for at least 1 min, and then washed with sterile distilled water 3~5 times. The MG was inoculated on EC induction medium after peeling off the seed coat. Method 2: The seeds from the cones were sterilized with 75% alcohol for 1~2 min on a super-clean bench, and washed with sterile distilled water 3~5 times. Then, the seeds were soaked in a 10% sodium hypochlorite solution for 15 min. After rinsing 3~5 times with sterile distilled water and peeling off the seed coat, the seeds were inoculated on EC induction medium with MGs as explants. The EC induction medium was the mLV medium [26] containing 4.0 g $L^{-1}$ gellan gum (Gel, Sigma-Aldrich, St Louis, MO, USA), 0.5 g $L^{-1}$ glutamine (Gln), 30 g $L^{-1}$ sucrose, 0.5 g $L^{-1}$ acid hydrolyzed casein, 2.0 mg $L^{-1}$ NAA, 1.5 mg $L^{-1}$ 6-BA, and 5 MGs (diameter of 90 mm and depth of 20 mm) per dish.

The culture media and culture conditions were the same in the following 5 parts.

MGs were used as the explants in the studies of the effects of cone cryopreservation at 4 °C, basal medium, combination of PGRs, ethylene removal, and liquid medium supplementation. The MG (Figure 1a) and ZE (Figure 1b) of the #11, #12 and #13 families were used as explants. Each dish contained 5 explants.

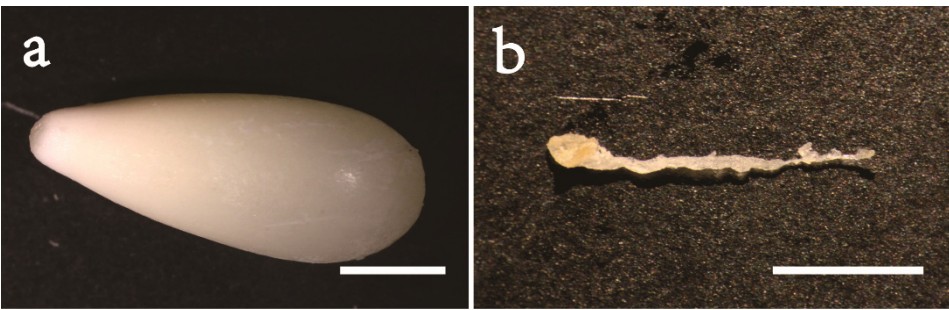

**Figure 1.** MG and ZE explants of Korean pine. (**a**) MG explants. Scale = 0.5 cm. (**b**) ZE explants. Scale = 0.5 cm.

### 2.2.2. Cold Storage of Cones at 4 °C

The MGs of the #1, #2, #3, #4 and #5 families were used as explants. One part of the cones was directly used for the EC induction test. The other part was wiped with alcohol cotton and stored at 4 °C for 28 days before the EC induction test; the MGs were used as explants, and the other culture conditions were the same as those in Section 2.2.1.

The morphological and structural changes of the Korean pine seeds before and after treatment were observed under a microscope (OLYMPUS SZX7, Tokyo, Japan; equipped with Canon DS126271 camera, Tokyo, Japan).

### 2.2.3. Basal Medium

The MGs of the #1, #2, #3, #4 and #5 families were used as explants. EC was induced in two basal media, mLV and DCR (the compositions of the two media are shown in Supplementary Table S1) [27]; the MGs were used as explants, and the other culture conditions were the same as those in Section 2.2.1.

### 2.2.4. Combination of PGRs

The MGs of the #11, #12 and #13 families were used as explants. The mLV basal medium was used for EC induction. There were two different hormone treatments: Treatment 1: NAA ($2.0$ mg $L^{-1}$) + 6-BA ($1.5$ mg $L^{-1}$) and Treatment 2: 2, 4-D ($5.0$ mg $L^{-1}$) + 6-BA ($1.5$ mg $L^{-1}$). The MGs were used as explants, and the other culture conditions were the same as those in Section 2.2.1.

### 2.2.5. Ethylene Removal and Liquid Medium Supplementation

The MGs of the #11, #12 and #13 families were used as explants. Three treatments were tested in this section. Treatment 1: EC induction medium was the same as that in Section 2.2.1. Treatment 2: 1 mL liquid induction medium (same as that in Section 2.2.1, but without Gel) was supplemented after 15 days of inoculation, and the culture dish was immediately sealed with plastic wrap. Treatment 3: after 15 days of inoculation, the Petri dish was opened for 1 h to fully release the ethylene, and then sealed with plastic wrap to continue the culture.

### 2.3. Optimization of Proliferation and Maturation Conditions for Korean Pine EC

When the diameter of the EC was 1~2 cm, the proliferation culture was started. The mLV medium containing 30 g $L^{-1}$ sucrose, 0.5 mg $L^{-1}$ 2,4-D, 0.1 mg $L^{-1}$ 6-BA, 0.5 g $L^{-1}$ acid hydrolyzed casein, 0.5 g $L^{-1}$ Gln, and 4.0 g $L^{-1}$ Gel served as the medium for the proliferation of Korean pine. Subculture was performed every two weeks, and the maturation test was carried out after 8 weeks.

Two cell lines, #001-001 and #001-100, were selected. The mLV medium was supplemented with 4.0 g $L^{-1}$ Gel, 0.5 g $L^{-1}$ Gln, 0.5 g $L^{-1}$ acid hydrolyzed casein, 1.0 mg $L^{-1}$ 2,4-D, and 0.5 mg $L^{-1}$ 6-BA. For sugar treatment, 30 g $L^{-1}$ sucrose or 30 g $L^{-1}$ maltose was used. Subculture was performed once every two weeks. In each subculture, 0.2 g EC was weighed and transferred to the same fresh proliferation medium. The fresh weight of the EC after proliferation was measured after 4 subculture cycles, and the maturation ability of the EC proliferated under sucrose and maltose conditions was determined.

Maturation test: The EC (80 mg) was transferred into a 50 mL centrifuge tube and then the liquid proliferation medium without PGRs was added. The tube was shaken vigorously for even dispersion. The mixture was transferred to filter paper using a 5 mL pipette, and filtered with a Buchner funnel. The filter paper with EC was placed on solid medium. The maturation medium was the mLV medium supplemented with 10 g $L^{-1}$ Gel, 0.5 g $L^{-1}$ Gln, 68 g $L^{-1}$ sucrose, 80 μmol $L^{-1}$ ABA, and 0.5 g $L^{-1}$ acid hydrolyzed casein. Each treatment was repeated 10 times and cultured at $23 \pm 2$ °C in the dark for 3 months. The number of somatic embryos was recorded.

The pH of all the media in this study was 5.8. After pH adjustment, the media were sterilized by high temperature and high pressure (121 °C, 20 min). Gln and ABA (0.22 μm)

were added after the media were cooled to about 55~60 °C. MGs were used as explants. After inoculation, they were cultured in the dark at 23 ± 2 °C.

Somatic embryo development observation: Fresh EC was placed on a slide, stained with 0.1% safranin for 10 min, and sealed with a coverslip. After even dispersion of the plant tissue, the slides were immediately observed using an optical microscope (OLYMPUS CX 31, Tokyo, Japan; equipped with Canon DS126271 camera, Tokyo, Japan).

### 2.4. Cryopreservation

Eleven proliferation cell lines of Korean pine were studied.

Proliferation culture method: The optimal proliferation conditions in Section 2.3 were used for the proliferation culture of the different cell lines. The initial inoculation amount of the different cell lines was 0.2 g. The EC was weighed after 14 days of proliferation culture, and there were 3 replicates per treatment.

Meanwhile, 11 cell lines were subjected to the cryopreservation test, using the previously described method by our group [8]. Specifically, after proliferation and culturing for 7 days, 3 g of EC with a good growth status and transparent filament was placed in a medium containing 12 mL of 0.4 mol $L^{-1}$ sorbitol and shaken (120 rpm $min^{-1}$) at 25 °C for 18 h in the dark. Then, 1.125 mL of 99% dimethyl sulfoxide, 1.125 mL of 0.8 mol $L^{-1}$ sorbitol medium, and 0.75 mL of 0.4 mol $L^{-1}$ sorbitol medium were added and incubated on ice for 1.5 h. After that, the mixture was transferred to a 1 mL centrifuge tube, which was then placed in a programmed cooling box, quickly stored in a −80 °C refrigerator for 2 h, and then quickly stored in liquid nitrogen.

The ECs of the 11 cell lines were cultured to explore whether there are differences between the differentiation and maturation ability of EC somatic embryos without cryopreservation and after cryopreservation. The centrifuge tube containing the EC of each cell line was quickly removed from the liquid nitrogen and placed in a 37 °C water bath for 2~3 min. After thawing, the EC mixture was removed with a pipette, the surface liquid was sucked off, and the recovery culture was carried out. After 1 day, the EC was transferred to fresh proliferation medium. After 10 days, obvious new ECs were observed in different cell lines, and a large number of proliferated ECs were observed after 15 days, indicating that the cells could be restored after cryopreservation.

### 2.5. Statistical Analysis

The experimental data were statistically analyzed using Excel 2003, and the average value, induction percentage (%), proliferation efficiency (%), and maturation capacity (The number of somatic embryos per gram) were calculated according to the obtained data. One-way ANOVA and mean comparisons among treatments by the LSD method (at alpha = 0.05) were performed using SPSS 19 (IBM, New York, NY, USA). The graphs were plotted using Sigma-Plot 12.0 (Systat, Chicago, IL, USA).

## 3. Results

### 3.1. EC Induction of Korean Pine

#### 3.1.1. Sterilization Method of Explants

There was a significant difference in EC induction percentages among the different families ($p < 0.05$) (Table 1). The #10 family had the highest EC induction percentage, followed by the #8, and the #9 family had the lowest EC induction percentage. The method of complete sterilization of cones had a significant effect on the EC induction percentage of three Korean pine families (Table 1) ($p < 0.05$). Among them, the EC induction percentage of the #10 family increased the most. The EC induction percentage was 20.0% using the sodium hypochlorite sterilization method and 47.4% using the complete sterilization of the cones. The induction percentage increased by 2.4 times.

**Table 1.** Effects of sterilization methods of cones on EC induction percentage of Korean pine.

| Family | Complete Sterilization of Cones | | Sterilization with Sodium Hypochlorite | |
|--------|--------------------------------|--|----------------------------------------|--|
| | Total Explants | EC Induction Percentage (%) | Total Explants | EC Induction Percentage (%) |
| #8 | 102 | 23.5 b A | 70 | 14.3 ab B |
| #9 | 100 | 22.0 b A | 100 | 12.0 b B |
| #10 | 57 | 47.4 a A | 115 | 20.0 a B |

Note: Different lowercase letters in the same column and different uppercase letters in the same row indicate significant differences ($p < 0.05$).

3.1.2. Cryopreservation of Cones at 4 °C

The explants preserved at 4 °C for different amounts of time are shown in Figure 2. We found that the explants preserved at 4 °C for 0 days had abundant water contents (Figure 2a,b), while the explants stored at 4 °C for 28 days lost a lot of water and the surface was dry (Figure 2c,d). The moisture content of MGs (Figure 2b) without 4 °C cryopreservation was significantly higher than that of those stored at 4 °C for 28 days. (Figure 2d).

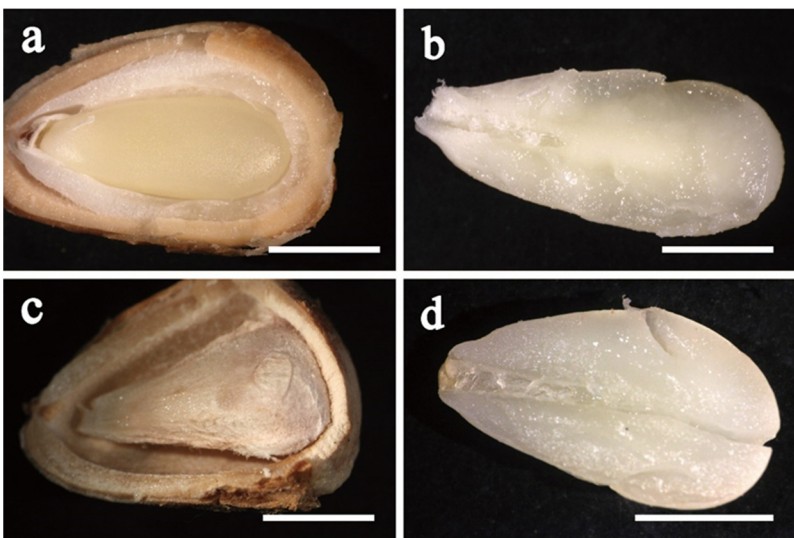

**Figure 2.** State of kernels and explants of immature seeds of Korean pine before and after 4 °C preservation. (**a**) The state of immature seeds preserved at 4 °C for 0 days. Scale = 1 cm. (**b**) The development status of immature ZEs preserved at 4 °C for 0 day. Scale = 0.6 cm. (**c**) The state of immature seeds preserved at 4 °C for 28 days. Scale = 1 cm. (**d**) The developmental state of immature ZEs preserved at 4 °C for 28 days. Scale = 0.6 cm.

Cold storage at 4 °C had a significant effect on the EC induction percentage of Korean pine ($p < 0.05$) (Table 2), and the EC induction percentages of the different families were also significantly different ($p < 0.05$). The EC induction percentage was 0 in two (#6 and #7) of the seven families. The EC induction percentage of the #2 family was basically unchanged after 4 °C preservation. The EC induction percentage was 4.1% without 4 °C preservation, and 4.0% after 28 days of 4 °C preservation. The EC induction percentage of the other four families increased to different degrees after 28 days of 4 °C preservation. Among them, the EC induction percentage of the #5 family without 4 °C preservation was 10.0%. After 28 days of 4 °C preservation, the EC induction percentage of the #5 family increased to 62.7%, an induction percentage increase of 5.3 times.

**Table 2.** Effects of cone pretreatment on EC induction percentage of Korean pine.

| Family | Preservation at 4 °C for 0 Days | | Preservation at 4 °C for 28 Days | |
|---|---|---|---|---|
| | Total Explants | EC Induction Percentage (%) | Total Explants | EC Induction Percentage (%) |
| #1 | 145 | 15.2 a A | 100 | 21.0 b A |
| #2 | 121 | 4.1 bc A | 50 | 4.0 c A |
| #3 | 60 | 1. 7 bc A | 148 | 6.1 c A |
| #4 | 70 | 12.9 a A | 85 | 23.5 b A |
| #5 | 10 | 10.0 ab A | 110 | 62.7 a B |

Note: Different lowercase letters in the same column and different uppercase letters in the same row indicate significant differences ($p < 0.05$).

### 3.1.3. Basal Medium

The EC induction percentages of the different families were significantly different ($p < 0.05$), and the EC induction percentage of the #5 family was the highest, with an induction percentage of 60.9%. The effects of DCR and mLV as the basal medium on the induction percentage of EC were also significantly different (Table 3) ($p < 0.05$). The induction percentages of ECs in the five families with DCR as the basal medium were higher than those with mLV as the basal medium. Among them, the induction percentage of the #5 family with mLV as the basal medium was 23.3%, while with DCR as the basal medium it was 60.9%, an increase of 2.6 times.

**Table 3.** Effect of basal medium on EC induction of Korean pine.

| Family | DCR | | mLV | |
|---|---|---|---|---|
| | Total Explants | EC Induction Percentage (%) | Total Explants | EC Induction Percentage (%) |
| #1 | 100 | 23.0 b A | 60 | 16.7 a A |
| #2 | 50 | 6.0 c A | 80 | 0 b B |
| #3 | 148 | 5.4 c A | 60 | 0 b A |
| #4 | 85 | 22.4 b A | 75 | 0 b B |
| #5 | 110 | 60.9 a A | 60 | 23.3 a B |

Note: Different lowercase letters in the same column and different uppercase letters in the same row indicate significant differences ($p < 0.05$).

### 3.1.4. Different Hormone Combinations

There was no significant difference in EC induction percentages among the different families (Table 4) ($p > 0.05$), while the effects of different PGRs on EC induction percentages were significantly different ($p < 0.05$). The results showed that the combination of 2,4-D and 6-BA was better than that of NAA and 6-BA, which was beneficial to EC induction, and the highest EC induction percentage (16.3%) was found in the #12 family.

**Table 4.** Effects of hormone combinations on EC induction percentage of Korean pine.

| Family | 2,4-D + 6-BA | | NAA + 6-BA | |
|---|---|---|---|---|
| | Total Explants | EC Induction Percentage (%) | Total Explants | EC Induction Percentage (%) |
| #11 | 120 | 11.7 a A | 90 | 8.9 a A |
| #12 | 160 | 16.3 a A | 100 | 8.0 a B |
| #13 | 80 | 10.0 a A | 65 | 9.2 a A |

Note: Different lowercase letters in the same column and different uppercase letters in the same row indicate significant differences ($p < 0.05$).

### 3.1.5. Liquid Medium Supplementation and Ethylene Removal

Liquid induction medium supplementation and ethylene removal after 15 days of EC induction significantly increased the induction percentage of EC ($p < 0.05$) (Table 5). After 15 days of induction culture, the liquid induction medium was added, and the induction

percentages of ECs in three families were increased to varying degrees. The induction percentage of ECs in the #12 family was 14.6%, and in the control, it was 8.0%. After 15 days of induction culture, the induction percentages of ECs in the #11 and #12 families were significantly increased after ethylene removal, and in the #13 family it was decreased (2.5%). The induction percentage of ECs in the control was 8.9%.

**Table 5.** Effects of ethylene removal and liquid medium supplementation on EC induction percentage of Korean pine.

| Family | Control | | Liquid Medium Supplementation | | Ethylene Removal | |
|---|---|---|---|---|---|---|
| | Total Explants | EC Induction Percentage (%) | Total Explants | EC Induction Percentage (%) | Total Explants | EC Induction Percentage (%) |
| #11 | 60 | 8.3 a B | 40 | 12.5 a AB | 40 | 17.5 a A |
| #12 | 125 | 8.0 a B | 55 | 14.6 a AB | 40 | 20.0 a A |
| #13 | 90 | 8.9 a A | 50 | 10.0 a A | 40 | 2.5 b B |

Note: Different lowercase letters in the same column and different uppercase letters in the same row indicate significant differences ($p < 0.05$).

### 3.1.6. Explant Sampling Site

Both MGs (Figure 3a) and ZEs (Figure 3b) could induce EC. The effect of explant sampling site on EC induction percentage of Korean pine was not significant ($p > 0.05$) (Table 6). However, the difference in the EC induction percentages among the different families was significant (#11 family > #12 family). The EC induction percentage of ZE explants was higher than that of MG explants. The EC induction percentages of three families were improved to varying degrees, but the effect of explants on the EC induction percentage of Korean pine was not significant ($p > 0.05$). The EC induction percentage was 31.7% using ZEs as explants, which was the highest, while it was 26.2% using MGs as explants.

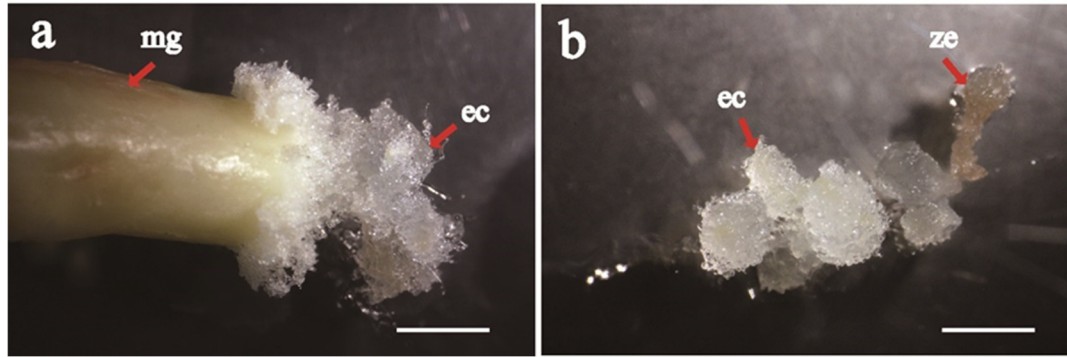

**Figure 3.** EC induced from MGs and ZEs as explants of Korean pine. (**a**) EC induced from MG explants. Scale = 1 cm. (**b**) EC induced from ZE explants. Scale = 1 cm. The letters ec stand for embryogenic callus; the letters mg stand for megagametophyte; and the letters ze stand for zygotic embryo.

**Table 6.** Effects of explant types on EC induction percentage of Korean Pine.

| Family | MG | | ZE | |
|---|---|---|---|---|
| | Total Explants | EC Induction Percentage (%) | Total Explants | EC Induction Percentage (%) |
| #11 | 65 | 26.2 a | 60 | 31.7 a |
| #12 | 55 | 12.7 b | 55 | 14.6 b |
| #13 | 50 | 4.0 b | 50 | 10.0 b |

Note: Different lowercase letters in the same column indicate significant differences ($p < 0.05$).

### 3.2. Effect of Maltose and Sucrose on Proliferation and Maturation

The proliferation efficiency of EC with sucrose was slightly higher than that with maltose, but the one-way ANOVA showed that maltose and sucrose had no significant effect on the proliferation efficiency of the ECs (Table 7) ($p > 0.05$). However, in terms of cell morphology, when maltose was used as a carbohydrate source, the number of early somatic embryos in the EC was higher and the cell walls were clearly delineated (Figure 4a). When sucrose was used as a carbon source, the number of early somatic embryos in EC was less and the cell walls were not clearly delineated (Figure 4b).

**Table 7.** Effects of maltose and sucrose on EC proliferation efficiency of Korean pine.

| Carbohydrate Type | Cell Lines | Proliferation Efficiency (%) |
|---|---|---|
| Maltose | #001-100 | 521.9 ± 44.7 a |
| Maltose | #001-001 | 595.5 ± 44.1 a |
| Sucrose | #001-100 | 714.4 ± 51.0 a |
| Sucrose | #001-001 | 678.0 ± 66.6 a |

Note: Different letters in the same column indicate significant differences ($p < 0.05$).

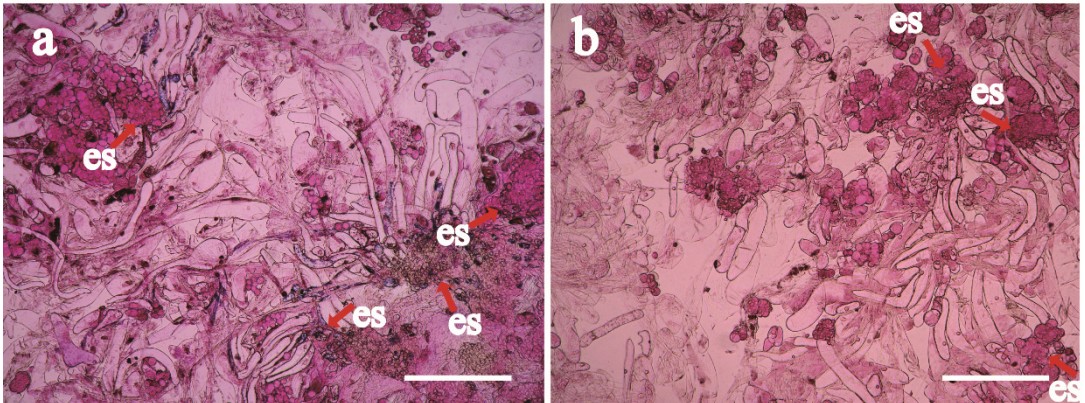

**Figure 4.** Effects of maltose and sucrose as carbohydrate sources on the development and maturation of Korean pine somatic embryos. (**a**) The proliferation of ECs cultured with maltose as a carbon source. Scale = 50 μm. (**b**) The proliferation of ECs cultured with sucrose as a carbon source. Scale = 50 μm. The letters es represent early somatic embryos.

The number of somatic embryos per gram of EC in the #001-100 cell line with maltose as a carbon source was higher than that with sucrose as a carbon source. The number of somatic embryos per gram of EC was 170.0 g$^{-1}$ FW with maltose as a carbon source (Table 8), while it was 143.8 g$^{-1}$ FW with sucrose as a carbon source. However, there was no significant difference between them ($p > 0.05$). The number of somatic embryos per gram of EC in the #001-001 cell line with maltose as a carbon source was lower than that with sucrose as a carbon source. The number of somatic embryos per gram of EC was 131.3 g$^{-1}$ FW when maltose was used as a carbon source, and 135.0 g$^{-1}$ FW when sucrose was used as a carbon source. The variance analysis showed no significant difference between maltose and sucrose ($p > 0.05$).

**Table 8.** Effects of maltose and sucrose on the development and maturation of Korean pine somatic embryos.

| Carbohydrate Type | Number of Somatic Embryos (No. g$^{-1}$ FW) | |
|---|---|---|
| | #001-100 Cell Lines | #001-001 Cell Lines |
| Maltose | 170.0 ± 16.5 a | 131.3 ± 17.0 a |
| Sucrose | 143.8 ± 20.0 a | 135.0 ± 18.1 a |

Note: Different letters in the same column indicate significant differences ($p < 0.05$).

*3.3. Cryopreservation*

Among the 11 cell lines, 4 cell lines did not have the ability to mature (Table 9), and the remaining 7 cell lines had the ability to form somatic embryos. The 11 cell lines were derived from the #1, #2 and #3 families. It can be seen that although a full-sib family was selected, the EC maturation ability induced by the same family was different, which may be due to the variation between cell lines. Among them, the #001-100 cell line had the best maturation ability. The number of somatic embryos per gram of EC was 168.8 g FW before cryopreservation and 160.0 $g^{-1}$ FW after cryopreservation. There was no significant difference before and after cryopreservation. The #001-001 cell line had the second-best maturation ability, with an somatic embryo number per gram of EC of 50.0 $g^{-1}$ FW before cryopreservation and 51.3 $g^{-1}$ FW after cryopreservation. There was little difference before and after cryopreservation. Meanwhile, the other nine cell lines had little difference in maturation ability before and after cryopreservation.

**Table 9.** Effect of cryopreservation on maturation of Korean pine somatic embryos.

| Cell Line | Before Cryopreservation (No. $g^{-1}$ FW) | After Cryopreservation (No. $g^{-1}$ FW) |
| --- | --- | --- |
| #001-001 | 50.0 ± 5.6 | 51.3 ± 9.0 |
| #001-017 | 0 | 10.0 ± 6.4 |
| #001-002 | 0 | 0 |
| #001-010 | 1.5 ± 0.4 | 13.8 ± 4.4 |
| #001-100 | 168.8 ± 13.1 | 160.0 ± 18.8 |
| #002-013 | 32.3 ± 9.4 | 25.0 ± 7.9 |
| #002-015 | 32.5 ± 8.6 | 30.0 ± 6.8 |
| #002-016 | 0 | 0 |
| #002-027 | 0 | 0 |
| #003-013 | 1.3 ± 1.2 | 0 |
| #003-014 | 0 | 0 |

## 4. Discussion

Reeves et al. have shown that sterilization of explants can significantly affect the EC induction rate, and sterilization of intact immature cones is sufficient to prevent contamination of the culture after inoculation and increase the EC induction rate [10]. This study also obtained a similar conclusion in that the method of complete sterilization of cones had a significant effect on the EC induction rate of three Korean pine families. Among them, the induction rate of ECs in the #10 cell line was the highest. The induction rate of ECs by sodium hypochlorite sterilization was 20.0%, while the induction rate of EC by the complete sterilization of the cones was 47.4%, which was 1.4 times higher. This study proposed a new sterilization method for Korean pine, which effectively improved the EC induction rate.

Traditionally, the EC induction rate is often improved by optimizing the site of the explants, the physiological state of the explants, and the culture conditions. Few studies have focused on the effect of cryopreservation of explants on EC induction rate. Low temperature treatment at 4 °C is used to promote SE of several angiosperms [28,29]. However, there are few reports on the effect of cryopreservation at 4 °C on pine SE. Hely et al. [30] demonstrated the beneficial effect of cryopreservation and opened the possibility of considering a cold preconditioning of plant materials as an alternative method to improve the somatic embryogenesis process in conifers. Montalbán et al. [31] found that the induction rate of EC increased when the cones were preserved at 4 °C for 1–2 months; however, long-term cryopreservation at 4 °C could decrease the induction rate of ECs. It was undetermined whether 4 °C cryopreservation of explants is beneficial to the EC induction rate of Korean pine. This study showed that due to the large size of Korean pine seeds, the water consumption was large during 4 °C cryopreservation. The longest preservation time was about 28 days, and a preservation time longer than 28 days may lead to dehydration

and death of immature seeds. The EC induction percentage of the #2 family was basically unchanged, while the EC induction percentage of the #5 family without 4 °C preservation was 10.00%. After 28 days of 4 °C preservation, the EC induction percentage of the #5 family increased to 62.73%, an induction percentage increase of 5.273 times. These results indicate the direction for future research.

Most studies have reported that the best explant for EC induction is ZE attached to the kernel [19,20]. The reason is that ZEs will absorb more growth regulators and nutrients from the embryo cavity [10,21], which will facilitate the initiation of embryonic cells. The study of Reeves et al. showed that the EC induction rate using ZEs as explants was significantly higher than that when using MGs as explants [10]. The EC induction rate using MGs as explants was very low, and the EC induction rate using ZEs as explants increased from 16.0% to 55.0% [10]. This study showed that the EC induction rate of ZE explants was higher than that of MG explants, and the EC induction rate of three families increased to varying degrees.

The EC is the basis of large-scale plant regeneration, an important material for genetic transformation, and an ideal system for studying cell differentiation and totipotency [32–35]. The quality of EC not only affects the proliferation efficiency of the EC, but also affects the quantity and quality of SEs [36,37]. The quality of somatic embryos is a key factor in evaluating the success of SEs. The studies on SEs of most conifer species have focused on the culture conditions for the maturation stage [38,39], such as basal medium, sugar type, gel concentration [40], etc. However, the culture conditions at the EC proliferation stage, which are crucial to the development and maturation of somatic embryos [31], have received little attention. During the proliferation of Douglas fir embryogenic tissues, the use of maltose in combination with Glitz, although not significant, generally resulted in better growth compared to sucrose as a carbohydrate source [10]. Gupta recommended the use of maltose for a wide range of conifers including Douglas fir, reporting that it resulted in the early-stage somatic embryos growing in size and vigor to advanced early-stage somatic embryos on a variety of media [41]. In Scots pine SEs, the size of the embryo increased when maltose was used instead of sucrose in MSG and DCR media [42]. This study found that the proliferation efficiency of EC with sucrose as a carbohydrate source was slightly higher than that with maltose, but the difference was not statistically significant. From the perspective of cell morphology, when maltose was used as a carbohydrate source, the number of early somatic embryos in the EC was higher and the cell walls was more clearly delineated. When sucrose was used as a carbon source, the number of early somatic embryos in EC was less and the cell walls were not clearly delineated. The number of somatic embryos per gram of EC with maltose as a carbon source was 170.0 $g^{-1}$ FW, which was higher than that with sucrose (143.8 $g^{-1}$ FW), but the difference was not statistically significant. Taken together, our results reveal that maltose is suitable as a carbon source for EC proliferation. Therefore, in the process of SE, not only the proliferation efficiency of EC, but also the maintenance time of the EC and the development and maturation ability of the somatic embryos should be considered. In breeding, SE and cryopreservation are combined to achieve large-scale reproduction of excellent plant resources [43,44]. The somatic embryo differentiation and maturation abilities will gradually weaken with the passage of culture time during the proliferation of conifer ECs [37].

Therefore, EC cryopreservation technology is of great significance [45]. After cryopreservation, the culture can be restored at any time when needed. Previous studies only focused on the survival of ECs after recovery culture [8]. Whether cryopreservation of ECs has an effect on the maturation ability of somatic embryos is rarely reported. However, the maturation ability of ECs is the key to the *in vitro* SE system and is the basis of large-scale breeding. For many conifer embryogenic tissues, cryopreservation protocols have been developed indicating their availability for long-term storage of such tissues. The most suitable cryopreservation protocol involves tissue pretreatment together with the use of an optimal combination of cryoprotectants [46]. Most common carbon source in the cryop-

reservation pretreatment of embryogenic cells are sugars (sucrose, maltose, and glucose) or sugar alcohols (mannitol and sorbitol). In this study, 11 cell lines of Korean pine ECs were used as test materials. The cryopreservation and recovery culture of Korean pine ECs were successfully achieved. The 11 cell lines were derived from the #1, #2 and #3 families. It can be seen that although this selection is a full-sib family, the EC maturation ability induced by the same family was different, which may be due to the variation between cell lines. Additionally, the EC maturation experiments found that there was no significant difference in EC maturation ability before and after cryopreservation, indicating that the cryopreservation method of this study is suitable for Korean pine, and has important significance for the large-scale breeding of Korean pine.

## 5. Conclusions

Complete sterilization of cones was beneficial to the induction of ECs, and ZE explants were better than MG explants. The induction rate of ECs was significantly increased after the collection of pine cones and cryopreservation at 4 °C for 28 days. The induction rate of ECs in the #5 family was 10.0% without cryopreservation at 4 °C, and increased to 62.8% after cryopreservation at 4 °C for 28 days. This study optimized the key technology of SE, and improved the EC induction rate and maturation ability of Korean pine. These results provide new experimental data for further optimizing the *in vitro* SE system in Korean pine.

**Supplementary Materials:** The following supporting information can be downloaded at: https://www.mdpi.com/article/10.3390/f14040850/s1, Table S1: The preparation methods of mother liquor of mLV and DCR medium.

**Author Contributions:** L.Y. and H.S. conceived and designed the study. F.G. collected plant materials and prepared SE samples for analysis. F.G. and L.Y. analyzed the results. F.G., Y.S. and R.W. contributed to the writing of the manuscript and data analyses. H.S., I.N.T. and A.M.N. revised the manuscript. All authors have read and agreed to the published version of the manuscript.

**Funding:** This research was funded by grants from the Innovation Project of the State Key Laboratory of Tree Genetics and Breeding (2021B01) and the National Key R&D Program of China (2017YFD0600600).

**Data Availability Statement:** The data presented in this study are available in this manuscript.

**Acknowledgments:** The authors would like to thank TopEdit (www.topeditsci.com, accessed on 9 January 2023) for its linguistic assistance during the preparation of this manuscript.

**Conflicts of Interest:** The authors declare no conflict of interest.

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
