# Peer review of "Optimization of Key Technologies for Induction of Embryogenic Callus and Maturation of Somatic Embryos in Korean Pine (Pinus koraiensis)"

_forests, doi:10.3390/f14040850_

Round 1
Reviewer 1 Report
Dear Editor in Chief of forests journal (Genetics and Molecular Biology section)
Hi and have a good day
I studied the article entitled ”Optimization of Key Technologies for Induction of Embryogenic Callus and Maturation of Somatic Embryos in Korean pine (Pinus koraiensis)” written by Gao et al. the material methods and results show the investigation was achieved in a good manner. However, I think merely emphasis on callus induction is a little poor manuscript so I recommend the writer of the article add the result about the regeneration of plants from embryogenic callus with pictures to improve the quality of the article. Also, other traits such as the fresh/dry weight of the callus, the volume of the callus, and other traits were added to the article if available for the authors. So I suggest a major revision for this article. Another correction for this MS is the following:
Abstract part:
Line 18: add two lines of introduction for this part.
Line 22: Zygotic embryo (ZE) was better than megagametophyte(MG) as explants. Is not better to write: The results showed Zygotic embryo (ZE) was better than megagametophyte(MG) as explants.
Line 27-28: Is not better to omit this sentence:
Among them, the induction rate of the 5# family cultured with the mLV basal medium was 23.33%, while that with the DCR basal medium was 60.91%, which increased by 2.61 times.
Line 34: what does it mean? Large-scale breeding technology
Introduction part:
It is true that someone has done this work and this work has not been done by any others, but the action mechanism of the treatments should also be mentioned in this part. For example, the action mechanism of cold, growth regulator, carbohydrates, etc., on EC. Especially if they are shown together with a diagram, it will be better to improve the article quality.
Material and methods part:
Line 93: 7 full-sib families: how were they produced? Please explain in three lines. What is the parental of full-sib mating?
Line 129: what is the composition of mLV basal and DCR? Please bring in a supplementary table.
-Is not their somaclonal variation for EC?
In material and methods: please explain the origin and difference between ZE and MG in three lines.
Line 198: The experimental data were statistically analyzed by Excel 2003. Is it true? If the answer is ok, please refer to which analysis?
Line 199-201: is better to write ===== One-way ANOVA and mean comparisons among treatments by LSD method (at alpha=0.05) were performed using SPSS 19 (IBM, USA).
Results part
Line 207: 10#family: what is family???
Table 1: where is the data of another family such as 1,2,3,4,5,6,7,11
Line 215: It is true: different uppercase letters in the same row indicate significant differences (p < 0.05). In all tables please correct this item.
Table 1: title of treatments does not correspond with the material and methods
Figure 1: what is the difference between b and d?
Table 2: the mean comparison (lowercase) for EC induction percentage (0 days) is not true. Please check and correct.
Table 3: lowercase letters were forgotten. Please control. And also the uppercase grouping is a mistake. Please check.
-Why different families for different traits were used? is it not better only use the same family for all parts of the MS?
Line 267: The induction percent-266 age increased by 81.88%. where is this data in table 5?
Figure 2: is not show the embryogenic callus of explants. Please bring another picture
Figure 3: is not clear
Table 6: the difference between the 12 and 13 families is high in EC induction percentage at MG; however, there is no difference. Please check.
Line 291: sucrose on
Table 9: mean comparison is not true. Please correct both column's upper and lowercase.
-did not investigate somaclonal variation in this research? It is very important for EC.
-please bring the regeneration data of plantlets from EC to improve the quality of the article.
Discussion part
In this part please refer to the action of maltose and also cryopreservation in EC. Only comparing your result with others is not beneficial.
Line370-372: please refer there is no significant difference
-why the authors used MG explant while the result showed ZE better than that.
Line 412: large-scale breeding of excellent plant resources==== what does it means?
Best wishes
Author Response
Dear reviewer,
Our sincere thanks to you for the time and effort that you have put into reviewing our manuscript! We found all the comments very constructive and helpful, and have revised our manuscript according to all comments.
Thank you for considering our revised manuscript!
Point 1: Line 18: add two lines of introduction for this part.
Response 1: Added:Somatic embryogenesis (SE), which leads to the formation of embryonic callus (EC) tissue, is the most promising method of large-scale production and selective breeding of woody plants. However, in many species, SE suffers from low induction and proliferation rates, hindering the production of improved plant materials.
Point 2: Line 22: Zygotic embryo (ZE) was better than megagametophyte (MG) as explants. Is not better to write: The results showed Zygotic embryo (ZE) was better than megagametophyte (MG) as explants.
Response 2: “Zygotic embryo (ZE) was better than megagametophyte(MG) as explants” has been changed to “The results showed Zygotic embryo (ZE) was better than megagametophyte(MG) as explants”。
Point 3: Line 27-28: Is not better to omit this sentence:
Response 3: Deleted
Point 4: Line 34: what does it mean? Large-scale breeding technology
Response 4: Deleted
Introduction part:
Point 5: It is true that someone has done this work and this work has not been done by any others, but the action mechanism of the treatments should also be mentioned in this part. For example, the action mechanism of cold, growth regulator, carbohydrates, etc., on EC. Especially if they are shown together with a diagram, it will be better to improve the article quality.
Response 5: The mechanism of action of growth regulators and carbohydrates on EC has been added.
Material and methods part:
Point 6: Line 93: 7 full-sib families: how were they produced? Please explain in three lines. What is the parental of full-sib mating?
Response 6: Modified. On July 1, 2018, the cones of 7 full sibling families (Numbered 1#(female parent*male parent: 176*174), 2#(female parent*male parent: 166*158), 3#(female parent*male parent: 176*175), 4#(female parent*male parent: 174*174), 5#(female parent*male parent: 175*161), 6#(female parent*male parent: 166*161), and 7#(female parent*male parent: 174*158) of Korean pine were obtained from Lushuihe Seed Orchard, Jilin Province, China.
Point 7: Line 129: what is the composition of mLV basal and DCR? Please bring in a supplementary table.
Response 7: Added.
Point 8: In material and methods: please explain the origin and difference between ZE and MG in three lines.
Response 8: ZE and MG tablets have been supplemented, as shown in Figure 1.
Point 9: Line 198: The experimental data were statistically analyzed by Excel 2003. Is it true? If the answer is ok, please refer to which analysis?
Response 9: Modified.
Point 10: Line 199-201: is better to write ===== One-way ANOVA and mean comparisons among treatments by LSD method (at alpha=0.05) were performed using SPSS 19 (IBM, USA).
Response 10: One-way ANOVA and mean comparisons among treatments by LSD method (at alpha=0.05) were performed using SPSS 19 (IBM, USA).
Results part
Point 11: Line 207: 10#family: what is family???
Response 11: Family refers to the pedigree. The experimental materials selected in this study come from different pedigree.
Point 12: Table 1: where is the data of another family such as 1,2,3,4,5,6,7,11
Response 12: In some experiments, we only selected 8 #, 9 # and 10 # families, without other families.
Point 13: Line 215: It is true: different uppercase letters in the same row indicate significant differences (p < 0.05). In all tables please correct this item.
Response 13: Modified.
Point 14: Table 1: title of treatments does not correspond with the material and methods
Response 14: Modified.
Point 15: Figure 1: what is the difference between b and d?
Response 15: Modified.
Point 16: Table 2: the mean comparison (lowercase) for EC induction percentage (0 days) is not true. Please check and correct.
Response 16: Modified.
Point 17: Table 3: lowercase letters were forgotten. Please control. And also the uppercase grouping is a mistake. Please check.
Response 17: Modified.
Point 18: -Why different families for different traits were used? is it not better only use the same family for all parts of the MS?
Response 18: Due to the high requirements for the development status of explants and the large demand for cones in this study, in the process of seed collection in 2018-2020, the number of cones collected in 3 years was different due to the incorrect development status of seeds and the insufficient number of cones in some families.
Point 19: Line 267: The induction percent-266 age increased by 81.88%. where is this data in table 5?
Response 19: Modified.
Point 20: Figure 2: is not show the embryogenic callus of explants. Please bring another picture
Response 20: Fig.2a is the EC grown from MG explants, and Fig.2b is the EC grown from ZE explants. The specific conditions of explants are shown in the newly added Fig.1.
Point 21: Figure 3: is not clear
Response 21: I have replaced the image with a 500ppi image.
Point 22: Table 6: the difference between the 12 and 13 families is high in EC induction percentage at MG; however, there is no difference. Please check.
Response 22: Modified.
Point 23: Table 9: mean comparison is not true. Please correct both column's upper and lowercase.
Response 23: Modified.
Point 24: -did not investigate somaclonal variation in this research? It is very important for EC.
Response 24: Added.
Point 25: -please bring the regeneration data of plantlets from EC to improve the quality of the article.
Response 25: Due to the epidemic, we did not obtain plant regeneration data in this part of the experiment.
Discussion part
Point 26: In this part please refer to the action of maltose and also cryopreservation in EC. Only comparing your result with others is not beneficial.
Response 26: The discussion on maltose and cryopreservation was supplemented.
Point 27: -why the authors used MG explant while the result showed ZE better than that.
Response 27: Because the experiment of ZE and MG was done in 2020, and we have been using MG as explants before 2020.
Point 28: Line 412: large-scale breeding of excellent plant resources==== what does it means?
Response 28: The results provide new experimental data for further optimizing the technical system of SE in Korean pine.

Reviewer 2 Report
Introduction
Line 79 This sentence belongs to the material and methods
Material and methods
Line 100-119 Are all genotypes included in this experiment? If not, please write which genotypes were tested. This applies to all research in chapter 2.2. (2.2.1, 2.2.2, 2.2.3, 2.2.4, 2.2.5).
Line 118 You mention Treatment 1 and Treatment 2. Please explain which treatment you mean.
Results
Line 213 You showed the results for only 3 genotypes. Where were the results for other genotypes? If only these genotypes are included in these treatments, you should emphasize the material and method. This applies to all results.
Line 237 You showed the results for only 7 genotypes. Where were the results for other genotypes?
Line 249 You showed the results for only 5 genotypes. Where were the results for other genotypes?
Line 258 You showed the results for only 3 genotypes. Where were the results for other genotypes?
Line 271 You showed the results for only 3 genotypes. Where were the results for other genotypes?
Line 288 You showed the results for only 3 genotypes. Where were the results for other genotypes?
Discussion
Line 369-371 Please avoid mentioning the number of results in the discussion part, especially if you already mention that in the Results section.
Author Response
Dear reviewer,
Our sincere thanks to you for the time and effort that you have put into reviewing our manuscript! We found all the comments very constructive and helpful, and have revised our manuscript according to all comments.
Thank you for considering our revised manuscript!
Point 1: Line 79 This sentence belongs to the material and methods
Response 1: The last paragraph of the introduction has been completely revised.
Material and methods
Point 2: Line 100-119 Are all genotypes included in this experiment? If not, please write which genotypes were tested. This applies to all research in chapter 2.2. (2.2.1, 2.2.2, 2.2.3, 2.2.4, 2.2.5).
Response 2: Modified
Point 3: Line 118 You mention Treatment 1 and Treatment 2. Please explain which treatment you mean.
Response 3: Modified
Results
Point 4: Line 213 You showed the results for only 3 genotypes. Where were the results for other genotypes? If only these genotypes are included in these treatments, you should emphasize the material and method. This applies to all results.
Response 4: Family information used has been described in materials and methods.
Point 5: Line 237 You showed the results for only 7 genotypes. Where were the results for other genotypes?
Response 5: Family information used has been described in materials and methods.
Point 6: Line 249 You showed the results for only 5 genotypes. Where were the results for other genotypes?
Response 6: Family information used has been described in materials and methods.
Point 7: Line 258 You showed the results for only 3 genotypes. Where were the results for other genotypes?
Response 7: Family information used has been described in materials and methods.
Point 8: Line 271 You showed the results for only 3 genotypes. Where were the results for other genotypes?
Response 8: Family information used has been described in materials and methods.
Point 9: Line 288 You showed the results for only 3 genotypes. Where were the results for other genotypes?
Response 9: Family information used has been described in materials and methods.
Discussion
Point 10: Line 369-371 Please avoid mentioning the number of results in the discussion part, especially if you already mention that in the Results section.
Response 10: Modified
Round 2
Reviewer 1 Report
Dear Editor in chief of Forest Journal
Hi and have a good day
I studied the revised MS and all of the items corrected into the article; however, the new items added into MS need grammar correction/native English edition.
In line 97: I think PGR2 must be deleted.
The name of family in the material and methods could be added in suitable place while the authors wrote all of them in the end of paragraph with grammar errors like use and used.
please authors control the chemical name of macro and micro elements in the media at supplementary file and also the they write the name of organic compounds not chemical formula.
In table 9: please control the class of data since there are many mistakes into the row and columns letter.
best wishes
Author Response
Dear reviewer,
Our sincere thanks to you for the time and effort that you have put into reviewing our manuscript! We found all the comments very constructive and helpful, and have revised our manuscript according to all comments.
Thank you for considering our revised manuscript!
Point 1: I studied the revised MS and all of the items corrected into the article; however, the new items added into MS need grammar correction/native English edition.
Response 1:Modified.
Point 2: In line 97: I think PGR2 must be deleted.
Response 2: Deleted.
Point 3: The name of family in the material and methods could be added in suitable place while the authors wrote all of them in the end of paragraph with grammar errors like use and used.
Response 3: Deleted.
Point 4: please authors control the chemical name of macro and micro elements in the media at supplementary file and also the they write the name of organic compounds not chemical formula.
Response 4:Modified.
Point 5: In table 9: please control the class of data since there are many mistakes into the row and columns letter.
Response 5: The results of variance analysis in Figure 9 have been deleted.